# Comparative analyses of parasites with a comprehensive database of genome-scale metabolic models

**Maureen A. Carey**[1,2]*, **Gregory L. Medlock**[3¤], **Michał Stolarczyk**[4,5], **William A. Petri, Jr.**[2], **Jennifer L. Guler**[2,4], **Jason A. Papin**[2,3,6]*

1 Department of Microbiology, Immunology, and Cancer Biology, University of Virginia School of Medicine, Charlottesville, Virginia, United States of America, 2 Division of Infectious Diseases and International Health, Department of Medicine, University of Virginia School of Medicine, Charlottesville, Virginia, United States of America, 3 Department of Biomedical Engineering, University of Virginia School of Medicine, Charlottesville, Virginia, United States of America, 4 Department of Biology, University of Virginia, Charlottesville, Virginia, United States of America, 5 Center for Public Health Genomics, University of Virginia School of Medicine, Charlottesville, Virginia, United States of America, 6 Department of Biochemistry & Molecular Genetics, University of Virginia School of Medicine, Charlottesville, Virginia, United States of America

¤ Current address: Vedanta Biosciences, Inc., Cambridge, Massachusetts, United States of America
* maureen.carey@mac.com (MAC); papin@virginia.edu (JP)

**Data Availability Statement:** All models and code are available at https://github.com/maureencarey/

## Abstract

Protozoan parasites cause diverse diseases with large global impacts. Research on the pathogenesis and biology of these organisms is limited by economic and experimental constraints. Accordingly, studies of one parasite are frequently extrapolated to infer knowledge about another parasite, across and within genera. Model *in vitro* or *in vivo* systems are frequently used to enhance experimental manipulability, but these systems generally use species related to, yet distinct from, the clinically relevant causal pathogen. Characterization of functional differences among parasite species is confined to *post hoc* or single target studies, limiting the utility of this extrapolation approach. To address this challenge and to accelerate parasitology research broadly, we present a functional comparative analysis of 192 genomes, representing every high-quality, publicly-available protozoan parasite genome including *Plasmodium*, *Toxoplasma*, *Cryptosporidium*, *Entamoeba*, *Trypanosoma*, *Leishmania*, *Giardia*, and other species. We generated an automated metabolic network reconstruction pipeline optimized for eukaryotic organisms. These metabolic network reconstructions serve as biochemical knowledgebases for each parasite, enabling qualitative and quantitative comparisons of metabolic behavior across parasites. We identified putative differences in gene essentiality and pathway utilization to facilitate the comparison of experimental findings and discovered that phylogeny is not the sole predictor of metabolic similarity. This knowledgebase represents the largest collection of genome-scale metabolic models for both pathogens and eukaryotes; with this resource, we can predict species-specific functions, contextualize experimental results, and optimize selection of experimental systems for fastidious species.

paradigm and archived as DOI: 10.5281/zenodo.5960209.

**Funding:** This work was supported by the National Institutes of Health (GM via T32LM012416; R21AI119881 to JG and JP; R37AI026649 and R01AI026649 to WP), the PhRMA Foundation (MC), the Bill and Melinda Gates Foundation (Grand Challenges Exploration Phase I grant OPP1211869 to GM), and seed funding from the University of Virginia's Engineering-in-Medicine program (MC, WP, and JP). National Insittutes of Health: https://www.nih.gov PhRMA Foundation: http://www.phrmafoundation.org Bill and Melinda Gates Foundation: https://www.gatesfoundation.org University of Virginia's Engineering-in-Medicine program: https://engineering.virginia.edu/eim The funders had no role in study design, data collection and analysis, decision to publish, or preparation of the manuscript.

**Competing interests:** The authors have declared that no competing interests exist.

## Author summary

Comparative genomics and phylogeny-based assumptions are useful approaches to generate predictions about cellular behavior for data-poor organisms, such as unculturable but clinically-relevant pathogens. Computational approaches, including metabolic modeling, can accelerate such comparisons. Genome-scale metabolic network models serve as a knowledgebase for an organism and enable rigorous and quantitative comparisons of disparate and sparse data, such as genomics and biochemical data, within and across species. Here, we generated a pipeline to create metabolic network models for 192 genomes from protozoan parasites, including the malaria parasite and organisms that cause diarrhea, African sleeping sickness, and leishmaniasis. Importantly, this pipeline was developed to propagate manual curation efforts from one model to others as manual curation remains the field's 'gold standard' for high-quality networks. We compare metabolic behavior across parasites to contextualize experimental results and compare metabolism. We identify which organisms are metabolically similar for the purpose of identifying experimental model systems and find that both metabolic niche and phylogeny influence metabolic similarity.

## Introduction

Malaria, African sleeping sickness, many diarrheal diseases, and leishmaniasis are all caused by eukaryotic single-celled parasites; these infections result in over one million deaths annually and contribute significantly to disability-adjusted life years [1–3]. In addition, human infectious and related parasites infect domestic and wild animals, resulting in a large reservoir of human pathogens and diseased animal population [4]. This combined global health burden makes parasitic diseases a top priority of many economic development and health advocacy groups [5–7]. However, effective prevention and treatment strategies are lacking. No widely-used, efficacious vaccine exists for any parasitic disease (*e.g.* [8–12]). Patients have limited treatment options because few drugs exist for many of these diseases, drug resistance is common, and many drugs have stage specificity (*e.g.* [13–15]). Thus, there is a pressing need for novel, effective therapeutics. Beyond the economic constraints associated with antimicrobial development [16,17], antiparasitic drug development is technically challenging for two primary reasons: these parasites are eukaryotes and they are challenging to manipulate *in vitro*.

As protozoa, these parasites share many more features with their eukaryotic host than prokaryotic pathogens do. Thus, antiparasitics must target the parasite while minimizing the effect on potentially similar host targets, similar to cancer therapeutics. Enzyme kinetics can be leveraged such that the drug targets the pathogen's weak points while remaining below the lethal dose for host [18] or drugs can synergize with the host immune response (*e.g.* [19,20]). Unique parasite features (*i.e.* signalling cascades as in [21] or plastid organelles as in [22]) can also be targeted once identified.

Drug target identification and validation are further complicated by experimental challenges associated with these parasites. Many of these organisms have no *in vitro* culture systems, such as *Plasmodium vivax* (malaria) and *Cryptosporidium hominis* (diarrheal disease), or *in vivo* model system, such as *Cryptosporidium meleagridis* (diarrheal disease). Some parasite species have additional unique biology and resultant experimental challenges hindering drug development, such as resistance to genetic modification. For example, *Plasmodium falciparum* (malaria) was considered refractory to genetic modification until recently [23,24]. *Entamoeba*

*histolytica* (diarrheal disease) has also been refractory to efficient genetic manipulation, and the genomes of *Leishmania* develop significant aneuploidy under selective pressure [25,26].

Although these challenges may be circumvented with new technology, the use of clinical samples, and reductionist approaches, little data exist relative to that which is available for most bacterial pathogens. Without adequate profiling data (genome-wide essentiality, growth profiling in diverse environmental conditions, etc.), we do not have the knowledge to rationally identify novel drug targets. Untargeted and unbiased screens of chemical compounds for antiparasitic effects have proven useful (if the parasite can be cultured, *e.g.* [27–32]), but this approach provides little information about mechanism of action or mechanisms of resistance development. Typical approaches to study drug resistance, such as evolving resistance to identify mutations in a drug's putative target, are not possible without a long-term culture system and a relatively well-annotated genome.

As a result of these difficulties (**Table 1**), data collected in one organism are frequently extrapolated to infer knowledge about another parasite, across and within genera (**Fig 1**). *Toxoplasma gondii* is frequently used as a model organism for other apicomplexa due to its genetic and biochemical manipulability [33–36]. Mouse models of malaria [37,38] and cryptosporidiosis [39,40] imperfectly represent the disease and/or use different species than the human pathogen. However, the modest characterization of functional differences among parasite species, beyond comparative genomics (*e.g.* [41–44]), limits the utility of this extrapolation-based approach, especially broadly among protozoa. Systematic assembly of existing knowledge about parasites and their predicted capabilities could greatly improve the extrapolation-based knowledge transfer by facilitating rigorous *in silico* comparison. Such systems biology approaches (*e.g.* genome-scale metabolic modeling) provide a framework to understand parasite genomes, highlight knowledge gaps, and generate data-driven hypotheses about parasite metabolism.

Genome-scale metabolic models are built from genomic data and by inferring function to complete or connect metabolic pathways; these models are supplemented with data from functional genetic and biochemical studies, representing our best understanding of an organism's

**Table 1. Summary of select parasitic diseases and their causal organism.** Parasites cause important human and animal diseases and have unique biological and experimental challenges that have made interpretation of *in vivo* and *in vitro* data challenging. Several examples are shown. Current treatments and associated observed drug resistance are noted. Many well-studied parasites remain refractory to genetic modification and/or still have poor genome annotation. 'Uncharacterized' genes were identified via EuPathDB searches for terms such as 'uncharacterized', 'putative', 'hypothetical', etc., for a representative strain. Because each database is heavily influenced by the respective scientific community, some databases such as CryptoDB do not use these terms because the function of so few genes have been validated in the *Cryptosporidium* parasites. Thus, the genomes of the *Cryptosporidium* parasites are mostly hypothetical and proposed functions are only putative; the reported percent of genome that is hypothetical is low for this reason (highlighted by an asterisk).

| Species | Disease | Treatable? | Drug Resistance? | Culturable? | Genetically tractable? | Percent of genome is 'hypothetical'? |
|---------|---------|------------|------------------|-------------|------------------------|--------------------------------------|
| *Trypanosoma brucei* | African sleeping sickness | yes | yes | yes | yes | 76.40% |
| *Babesia bovis* | babesiosis | yes | no | yes | yes | 72.00% |
| *Trypanosoma cruzi* | Chagas disease | yes | yes | yes | yes | 52.90% |
| *Cryptosporidium hominis* | diarrhea | no | - | no | no | 54.10% |
| *Cryptosporidium parvum* | diarrhea | no | - | yes | yes | 4.1%* |
| *Entamoeba histolytica* | diarrhea | yes | yes | yes | yes | 79.80% |
| *Giardia intestinalis* | diarrhea | yes | yes | yes | yes | 39.20% |
| *Naegleria fowleri* | encephalitis | yes | yes | yes | no | 31.70% |
| *Leishmania major* | leishmaniasis | yes | yes | yes | yes | 76.60% |
| *Plasmodium falciparum* | malaria | yes | yes | yes | yes | 37.60% |
| *Plasmodium vivax* | malaria | yes | yes | no | no | 43.50% |
| *Toxoplasma gondii* | toxoplasmosis | yes | yes | yes | yes | 56.20% |
| *Trichomonas varginalis* | trichomoniasis | yes | yes | yes | yes | 94.00% |

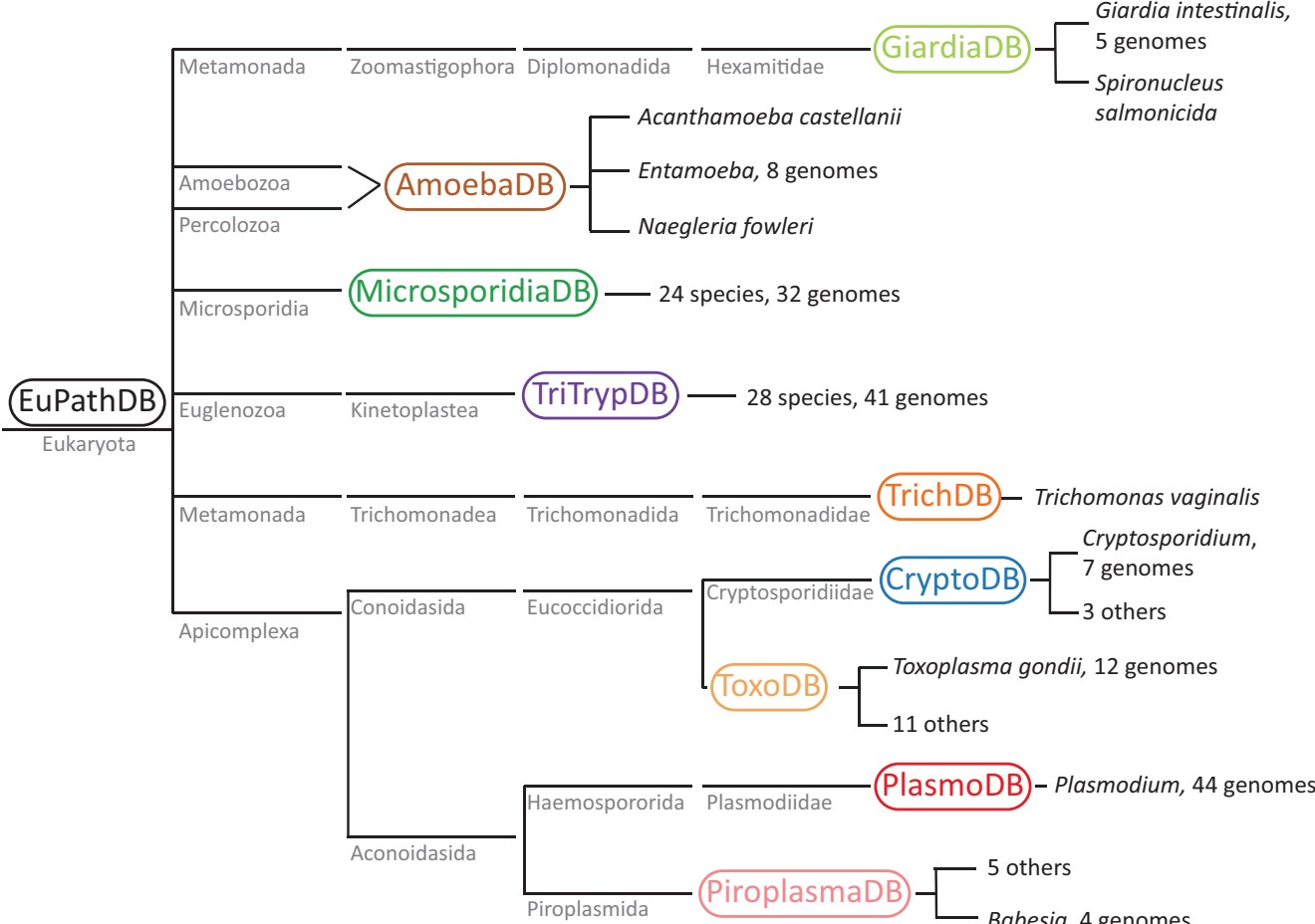

**Fig 1. EuPathDB databases.** EuPathDB is the Eukaryotic Pathogens database and serves as a repository for parasite 'omics data; EuPathDB contains field-specific databases including GiardiaDB, AmoebaDB, MicrosporidiaDB, TriTrypDB, TrichDB, CryptoDB, ToxoDB, PlasmoDB, and PiroplasmaDB (*all shown*), as well as FungiDB, HostDB, and MicrobiomeDB. Here, a phylogenetic tree of database member parasites is shown (lines are not to scale). Each EuPathDB sub-database is in a rough phylogenetic grouping, but the parasites on the EuPathDB databases are genetically and phenotypically highly diverse. Database color-coding shown here will be used through other figures.

biochemistry and cellular biology. Unfortunately, existing approaches for the construction of metabolic network models are lacking in standardization and scalability and/or biological relevance for eukaryotes. While there are pipelines that include compartmentalization (i.e. RAVEN [45] and merlin [46]), individual high-quality parasite reconstructions (e.g. [47–51]), and scalable pipelines for the construction of many networks (i.e. CarveMe [52], ModelSEED [53]), we sought to build on these tools and the Eukaryotic Pathogens Database (EuPathDB [54]) to leverage genomic information on the EuPathDB database and existing effort towards manual curation of individual reconstructions.

Here, we present a parasite knowledgebase, **Para**site **D**atabase **I**ncluding **G**enome-scale metabolic **M**odels (**ParaDIGM**), for this purpose. ParaDIGM is a collection of publicly available genome-scale metabolic models, and the computational tools needed to generate and re-generate these models iteratively as new data becomes available. Importantly, these tools also enable the propagation of experimental data collected in a manual curation to closely related organisms. The integration of this genomic and experimental evidence into genome-scale metabolic models enables direct comparison of predicted metabolic capabilities in specific

contexts, rather than the purely qualitative comparisons that can be performed with traditional genomic approaches. We demonstrate the utility of ParaDIGM by comparing metabolic capacity, gene essentiality, and pathway utilization. Ultimately, ParaDIGM can be used to better leverage experimentally tractable model systems for the study of eukaryotic parasites and antiparasitic drug development.

## Results

### Building ParaDIGM, a parasite knowledgebase

To build a comprehensive collection of genome-scale network reconstructions representing parasite metabolism, we designed a novel network reconstruction pipeline optimized for eukaryotic organisms (**Fig 2A**). Our pipeline builds on publicly available, open source software and resources [52,54–56] and focuses on the compartmentalization of biochemical reactions (**Fig 2A**). We applied this pipeline to assemble networks for all publicly available reference genomes from parasite isolates representing 119 species (see **Data Availability** for link to code and reconstructions). In brief, we obtained 192 high-quality genomes from the parasite genome resource, EuPathDB [54], to generate a *de novo* reconstruction for each genome (**Fig 2A**, *step 1*). We mapped the protein sequence of all open reading frames against a biochemical database [56] to identify putative metabolic functions via gene-protein-reaction mappings. Reaction compartmentalization was adjusted to maintain each gene-protein-reaction mapping but only with the subcellular compartments relevant for each organism. A large proportion of parasite gene-reaction pairs would otherwise be misassigned or removed from the network due to assignment to an incorrect compartment, due to lack of orthologous and compartmentalized reactions in biochemical databases; our pipeline reassigns these reactions to the cytosol or extracellular space (**Fig 2B**). Although not all functions annotated on EuPathDB are integrated into our *de novo* reconstructions using this approach, well studied enzymes and pathways are well represented (**S1 Fig**); discrepancies between EuPathDB and *de novo* reconstructions can be prioritized in future curation efforts. We also identify metabolic functions not currently annotated on EuPathDB (**S1A and S1B Fig**).

We next leveraged the manual curation in one parasite reconstruction, *P. falciparum* (**Fig 2A**, *step 2*, curation from [47,57] and in **S1 Table**), to generate a semi-curated reconstruction for a subset of phylogenetically-related organisms, specifically all *Plasmodium sp.*. To build these semi-curated reconstructions, we transformed the manually-curated reconstruction using genetic orthology (**Fig 2A**, *step 3*) and added all transformed reactions to the recipient *de novo* reconstruction (**Fig 2A**, *step 4*), improving the overlap between our curated and draft networks for *Plasmodium* reconstructions (**S2 Fig**). Lastly, all draft and semi-curated reconstructions were gapfilled using parsimonious flux balance analysis (pFBA)-based gapfilling [60,61] to complete biochemical requirements identified in the experimental literature (**Fig 2A**, *step 5*) and to produce biomass (see the *Additional Information*: *Online Methods*). Gapfilling too adds to the metabolic scope of all reconstructions (**S3 Fig**). As a result, when compared to manually-curated parasite reconstructions [47–50], semi-curated reconstructions are larger in scope than *de novo* reconstructions (**Fig 2C**) and generate predictions with comparable or improved accuracy (**Fig 2D**). These reconstructions are also more compliant with community standards [62,63] than previous reconstructions for parasites (representative MEMOTE examples shown at https://github.com/maureencarey/paradigm/tree/master/memote_reports).

Our *de novo* draft reconstructions contain only genetically supported information (prior to gapfilling) and, unsurprisingly, reconstruction size is correlated with genome size (**Fig 3A and 3B**). The large genome of *Chromera velia* CCMP2878 (a non-parasitic organism on CryptoDB with 31,799 ORFs and 3,064 reactions) corresponds to a reconstruction with the second most

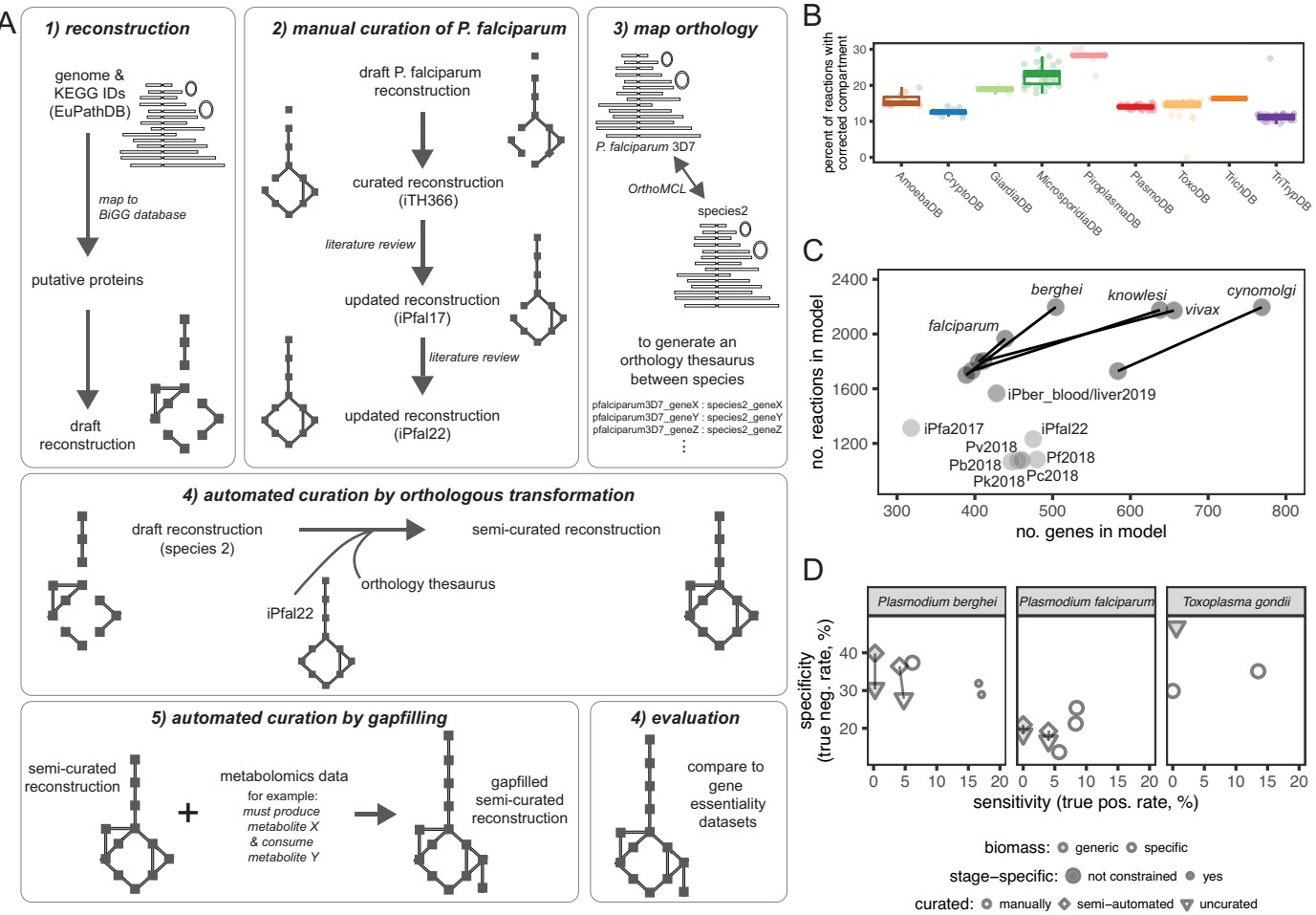

**Fig 2. Building a parasite knowledgebase.** Genetic data (from EuPathDB), orthology information (from EuPathDB's OrthoMCL), and biochemical data from metabolomics studies (acquired from a literature review) were used to build our reconstructions in a multistep process; gene essentiality data was used to evaluate resultant models. **(A): Reconstruction pipeline.** First, *de novo* reconstructions are built from annotated genomes and supplemented with KEGG reaction-associated genes on the database (see **Additional Information: Online Methods**). Next, we curated an existing manually curated reconstruction for *P. falciparum* 3D7. Third, we mapped orthologous genes so that (fourth) we could add all metabolic functions from our curated *iPfal22* into the *de novo* reconstruction by transforming each gene-protein-reaction rule via orthology. Lastly, we performed automated curation by gapfilling reconstructions to known metabolic capabilities and to generate biomass. With the resulting reconstruction, we can compare simulations to experimental data such as gene essentiality screens. **(B): Considering compartmentalization.** Our approach moves a large proportion of the reconstruction's reactions from compartments in a biochemical database to biologically-relevant compartments (*e.g.* periplasm to extracellular). Thus, our *de novo* reconstruction approach accounts for compartmentalization, unlike many previous metabolic network reconstruction pipelines. Each model is represented by a point. Boxplots for each database denote the interquartile range with the median value at center; whiskers extend to 1.5 times the inter-quartile range (i.e. distance between the first and third quartiles) above or below the median. **(C): Orthology adds information.** Orthology-based curation improves reconstruction scope regarding total number of genes and reactions. These semi-curated reconstructions (each labeled dark point) are larger in scope due to the addition of reactions associated with genes added via orthologous-transformation. Semi-curated reconstructions are connected via a line to the draft uncurated reconstruction for that genome. Semi-curated reconstructions are named by the associated species; *Plasmodium* species are labeled with species name. Light colored dots represent previously published *Plasmodium* reconstructions (iPfal22, from [57] and [47], iPfa2017 from [49], iPbe-blood and iPbe-liver from [58], all others from [48]). **(D): Prediction accuracy.** Semi-curated reconstructions (diamonds) recapitulate the biology of experimentally-facile parasites as well as published, manually-curated reconstructions (circles). We tested accuracy of model predictions from the *de novo* reconstruction (triangle) and the final orthology-translated and semi-curated reconstruction (diamond) for *P. berghei* and compared these summary statistics to the prediction accuracy generated by our well-curated *iPfal22* and other previously published reconstructions [48–50,58,59]. This comparison was used to motivate our approach over *de novo* reconstruction building as our pipeline generates a reconstruction with greater predictive accuracy than *de novo* reconstruction and comparable to a well-curated reconstruction.

unique reactions with 58. Unique reactions are defined here as reactions found in only one reconstruction and no other reconstructions. However, even small reconstructions contain unique reactions prior to gapfilling (**Figs 3A** and **S4A**) and the vast majority of these unique reactions are well connected within the network (**S4B and S4C Fig**). In fact, 39 reconstructions

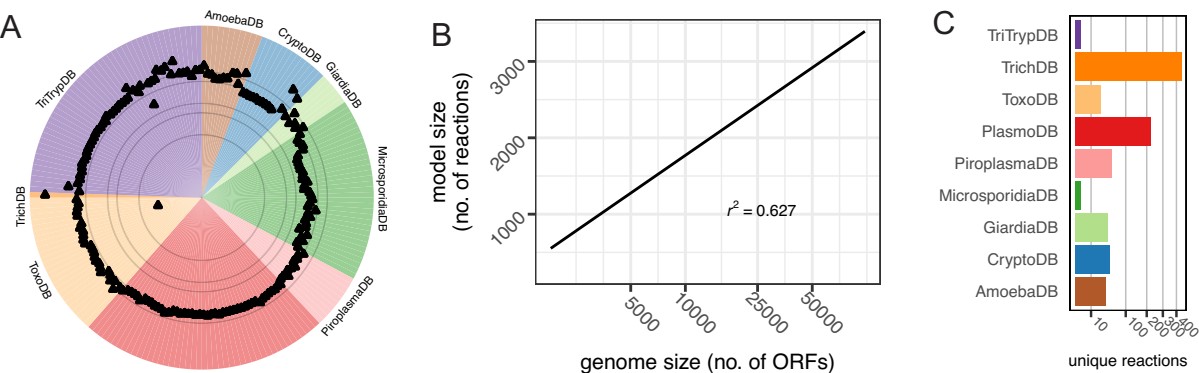

**Fig 3. Reconstructions for all eukaryotic organisms with published genomes. (A): Model summary.** Genome size is measured here by the number of amino acid sequences encoded by the genome (*triangle*) and model size is measured by the number of reactions present in the network (*square points*). Grey rings highlight 100, 500, 1000, 5000, and 10,000 ORFs moving from the center outwards. Genomes are grouped by database, a rough phylogenetic grouping (see **Fig 1**). Note: *T. gondii* RH is excluded from all future analyses given only a subset of the genome is available from EuPathDB. **(B): Model size is correlated with genome size.** Larger genomes tend to generate larger models. Line is fit to a linear regression with R2 noted (p-value < 0.001); the standard error is not shown. Points are color-coded by database. **(C): Unique reactions by database.** Number of unique metabolic reactions per database. Unique reactions are defined here as reactions found in every reconstruction within a database grouping and in no other reconstructions outside of that database grouping. Reactions found in different cellular compartments are considered distinct reactions.

contain at least one unique reaction (**S4A Fig**) and every database has unique functions, or functions that are found in every reconstruction within a database but not other reconstructions (**Fig 3C**). For example, the group of *Plasmodium* networks share over 200 reactions that are only found in *Plasmodium* reconstructions; among these reactions include hemoglobin breakdown (**Fig 3C**). The number of unique reactions is correlated with genome size, both before and after gapfilling (**S4F and S4G Fig**).

In sum, 34% of reactions are in fewer than 10% of models (**Fig 4**, *light grey*) and 352 reactions are unique to just a single model (examples in **Fig 4**). Importantly, these unique reactions are typically well-connected within the network and rarely represent blocked or unconnected reactions (**S4B and S4C Fig**). A core set of 45 reactions are contained in all 192 reconstructions (**Fig 4**). Just 3% of reactions are in at least 90% of models (*dark grey* in **Fig 4**); reactions shared

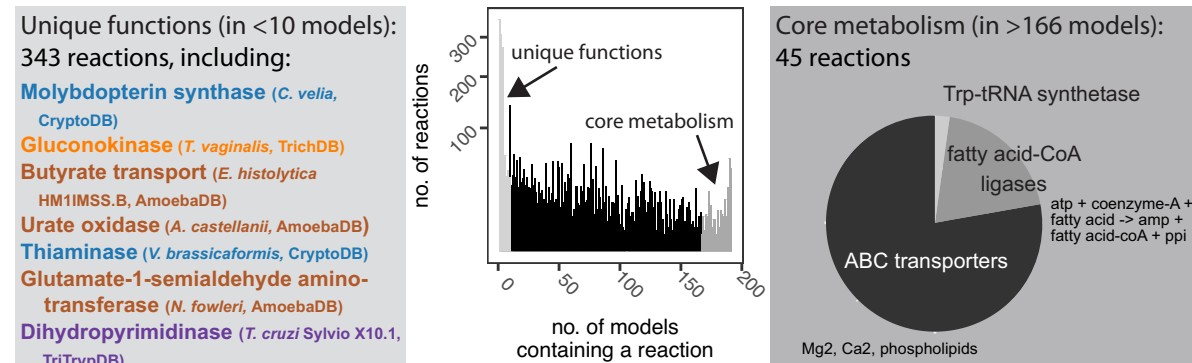

**Fig 4. Reaction frequency ranges from unique to core metabolism.** Reconstructions help identify rare metabolic functions (*light grey* box and on histogram, in fewer than 10 reconstructions) and core parasite metabolism (*dark grey* box and on histogram, in more than 166 reconstructions). Example rare reactions include seven metabolic reactions that are found in only one reconstruction. Of the 45 reactions found in all reconstructions (core metabolism), most reactions correspond to ABC transporter functions for ions or phospholipids. One reaction corresponds with a tRNA synthetase and the remaining correspond to fatty acid-CoA ligases for various fatty acids.

by many models include functions such as glycolytic enzymes. The relationship between genome size and model size is weakened following gapfilling (**S4D Fig**), likely due to the same biomass formulation for all reconstructions, and the frequency of rare reactions (*light grey* reactions in **Fig 4**) increases. ParaDIGM can be used to tease apart the difference between unique, species-specific functions and poorly annotated functions to illuminate the uncharacterized fraction of parasite genomes. To illustrate additional examples of using this resource, we identified niche-specific functions, predicted fluxomics studies to identify divergent enzymes, and identified representative model systems for drug development.

## Niche-specific metabolic functions

To identify niche-specific functions, we used ParaDIGM to compare the enzymatic capacity of each organism. Specifically, we compared which enzymes are genetically supported and, therefore, present in each reconstruction prior to gapfilling. We performed classical multidimensional scaling using the Euclidiean distance between reaction presence for each reconstruction (**Fig 5A and 5B**). We observe that phylogenetically-related parasites tend to contain similar reactions (**Fig 5A**). However, while networks generated from genomes within a common genera or species cluster together, models also cluster within environmental niche rather than broader phylogenetic grouping such as phylum. Apicomplexan parasites cluster tightly within genus but not across genera (**Fig 5A**, Apicomplexa colored by database). *Cryptosporidium* parasites cluster with other gut pathogens (**Fig 5A**, gut pathogens in *black*) rather than other Apicomplexa. Thus, phylogeny is not the sole predictor of model similarity (permutational multivariate analysis of variance, $p = 0.001$ using groups of *Cryptosporidium*, *Toxoplasma*, *Plasmodium*, and all other, and homogeneity of dispersion, $p < 0.001$).

Next, we performed random forest classification using reconstruction reaction content to identify the specific metabolic reactions associated with the metabolic niche of the gut environment. The classifier performed well with an AUC of 0.98 and an out-of-bag error rate of less than 8%, supporting our observation that gut parasites contain distinct metabolic reactions. Most important variables (reactions) were associated with being more frequently observed in non-gut pathogens, including gamma-glutamylcysteine synthetase (GLUCYS), glycerol-3-phosphate dehydrogenase (G3PD and G3PD4), an extracellular membrane proton pump (PPA_1), the glycine cleavage system (*GCCb*, GLYCL_2), phosphogluconate dehydrogenase (GND), and a pyruvate dehydrogenase using lipoamide (PDHa; **Fig 5C**). Reactions associated with gut pathogenicity included Butanal:NAD+ oxidoreductase (BNORhc), glucan 1-4-alpha-glucosidase (GLCGSD), and starch synthase (STARCH300Sc; **Fig 5C**).

Similarly, parasites that invade red blood cells, including *Plasmodium* and *Babesia*, are dissimilar when comparing their full reaction content (**Fig 5B**, *triangles*); however, the same analysis limited to each organism's genetically encoded transporters reveals that these parasites have relatively similar transporter capabilities (**Fig 5C**, *triangles*). This result indicates that these red blood cell-invading parasites rely on similar nutrients from their host red blood cell. On the contrary, the broad metabolic niche of extracellular growth yielded some outliers regarding enzymatic capacity and transporter profile (**Fig 5B** and **5C**, *circles*), likely due to the range of environments that parasites capable of extracellular growth encounter.

## Predicting metabolic function

Beyond the direct comparison of enzyme presence, we can use ParaDIGM to predict metabolic functions and the functional consequences of reaction presence and network connectivity. This approach augments the analysis beyond mere genetic comparisons: some enzymes may not be discovered in the genome despite being necessary to perform biochemical function

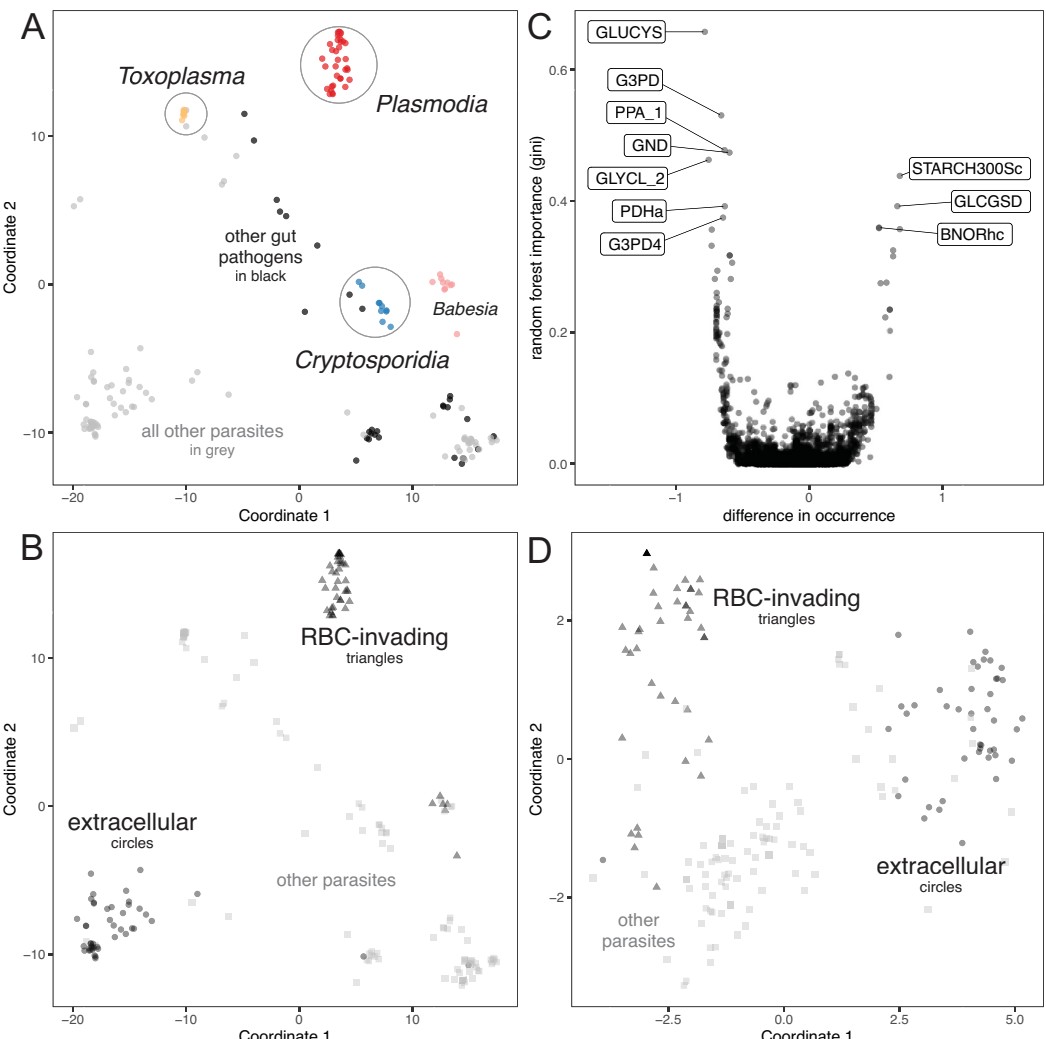

**Fig 5. Identifying metabolic niches. (A): Reaction content.** Classical multidimensional scaling was performed on the reaction content of all *de novo* reconstructions; each reconstruction is represented by a point (*grey/black* or colored by database for emphasis). Thus, this analysis focuses exclusively on the genetically supported features of each reconstruction. Apicomplexan parasites (colored by database) and all other gut pathogens (*black* points) are highlighted. **(B): Reaction content with alternative color scheme.** Parasites that invade red blood cells (*triangles*, *Plasmodium* and *Babesia*) or can replicate extracellularly (*circles*) are highlighted; all other parasites are in lighter grey squares. **(C): Important variables for the classification of gut pathogens.** We performed a random forest classification to distinguish organisms that are considered gut pathogens from other organisms in ParaDIGM (AUC = 0.98 and an out-of-bag error rate of less than 8%). Important variables with a difference in occurrence score of 1 were present in 100% of gut pathogens and 0% of other organism's reconstructions and those with a score of -1 were present in 100% of non-gut pathogens and 0% of gut pathogen's reconstructions. **(D): Transporter profile.** Again, parasites that invade red blood cells (*triangles*) or can replicate extracellularly (*circles*, like the kinetoplastids and *Giardia*, among others) are highlighted, with all other parasites are in lighter grey squares. Red blood cell-invading parasites cluster.

observed experimentally and are included in these models (**Fig 6A**). Relatively few fluxomics or controlled biochemical studies have been conducted for any one organism but these data can be predicted *in silico*. We simulate fluxomics studies to profile the metabolic capability of an organism using both genetic evidence and inferred network structure. To do this, we identified which metabolites can be consumed or produced in each model following gapfilling in a rich *in silico* media. This environment is simulated by permitting import of any metabolite for

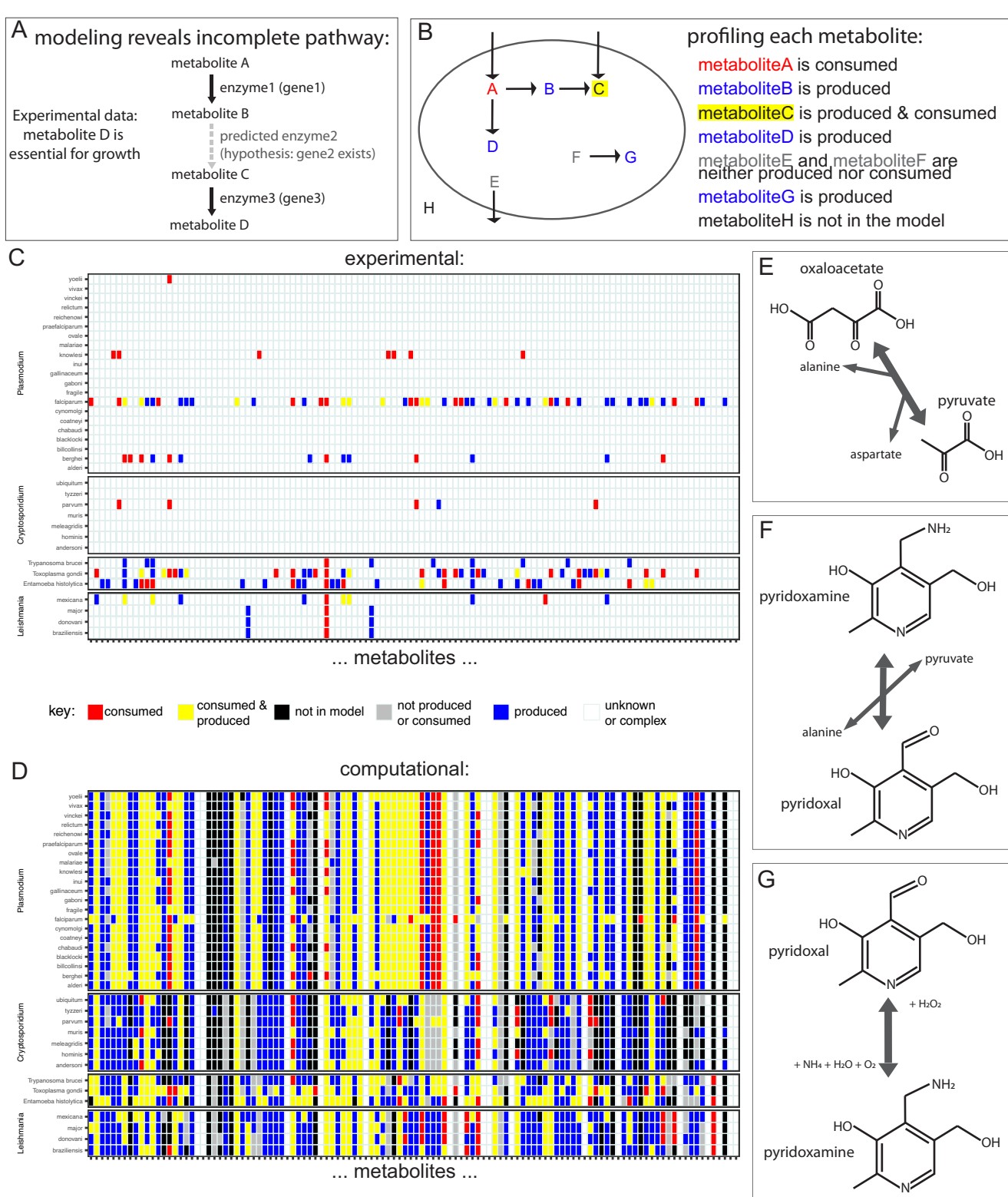

**Fig 6. Predicting metabolic function. (A): Advantage of network-based approaches.** Metabolic models include hypothetical functions (*i.e.* the enzyme encoded by *gene2*) that are unsupported by direct genetic evidence but may be indirectly required based on biochemical evidence. These functions are added through gapfilling. Using models augments our analysis beyond mere genetic comparisons: some enzymes may not be discovered in the genome despite being

necessary for biochemical observations made and are included in these models. **(B): Defining metabolic capacities.** With our gapfilled models, we can identify if metabolites are consumed and/or produced. **(C): Experimentally-derived metabolic functions.** We compiled data providing evidence for consumption or production of select metabolites from the literature (**S1 Table**). Consumed metabolites are imported by the parasite from the extracellular environment (*e.g.* the *in vitro* growth medium). Produced metabolites are synthesized by the parasite even when the metabolite is not in the extraceullar environment. See *Additional Information*: *Online Methods* for more detail. Data are sparse. **(D): Analogous *in silico* metabolic capacity.** Inferred metabolic capacity of each organism from Panel C for every metabolite from panel C. Data from panel C was used to gapfill reconstructions to generate data presented in Panel D (see **Fig 2A** for methods). See Panel B for definitions. Metabolites that are neither produced nor consumed are consumed intracellularly but are not taken up from the extracellular environment. Metabolites noted as 'complex or unknown' here are represented by multiple metabolite identifiers in the reconstructions (e.g., lactate is measured experimentally, but could represent both D-lactate and L-lactate within the reconstruction). **(E-G): Example gapfilled functions in the Vitamin B6 pathway.** These reactions were added to support the observed metabolic functions in Panel C or to support *in silico* growth. Panel E shows L-alanine-alpha-keto acid aminotransferase (*ASPTA6*, added to 58 reconstructions), Panel F shows pyridoxamine-pyruvic transaminase (*PDYXPT_c*, added to 64 reconstructions), and Panel G shows pyridoxamine oxidase (*PYDXO*, named pyridoxal oxidase in BiGG, added to 90 reconstructions). Note, a deaminateing pyridoxamine:oxygen oxidoreductase (*PYDXO_1*) is also added to 12 reactions to interconvert pyridoxal and pyridoxamine.

which there is a genetically-encoded transporter (**S2 Table**) or gapfilled transport reaction (**S3 Table**). A schematic for each metabolite categorization is shown in **Fig 6B** with experimental data shown in **Fig 6C**, with untested or unknown results in white, and analogous *in silico* results in **Fig 6D**. All models except for one (*Chromera velia* CCMP2878 with the largest genome) required gapfilling to synthesize one or more metabolites and/or biomass. We can expand the *in silico* predictions to all metabolites in all models (a total of 5,141 metabolites by 192 models, **S5 Fig**) to generate hypotheses about understudied metabolites and enzymes.

Interestingly, several metabolic enzymes were consistently predicted to be necessary for observed metabolic capabilities (metabolic tasks in **Fig 6C**) or growth across all parasites (gapfilled reactions in **Table 2**); three common examples are shown in **Fig 6E and 6G** including three steps in Vitamin B6 metabolism. Pyridoxamine oxidase (**Fig 6G**) is an understudied enzyme involved in Vitamin B6 metabolism; fewer than 300 articles on PubMed describe the enzyme. Not surprisingly given the lack of literature, the reaction associated with this enzyme is in just seven reconstructions in the BiGG database, including two iterations of the *S. cervisiae* S288C model [56]. The deaminating version of this reaction is in only 10 reconstructions in the BiGG database; all ten of these reconstructions are for eukaryotes including five *Plasmodium* genomes. Pyridoxamine oxidase was only added to the *V. brassicaformis* CCMP3155 and *G. niphandrodes* reconstructions in the bioinformatics-driven model construction steps; however, this enzyme was added in 90 gapfilling solutions to satisfy experimentally-derived functions. Thus, we predict that it is important for parasite growth. We also predict that the unidentified sequences for pyridoxal oxidase are highly divergent from known sequences because they were not identified using bioinformatic annotation methods. By comparing the reconstructions within ParaDIGM, we can identify high-confidence reactions that are encoded by divergent genetic sequences and missed by purely bioinformatic approaches.

## Selecting the most representative model system for an experiment

Genome-wide essentiality screens are available for *Plasmodium falciparum* [64] and *berghei* [58,65], *Toxoplasma gondii* [35], and *Trypanosoma brucei*. Using the models generated with ParaDIGM, we can perform the equivalent *in silico* simulations (**Fig 2D** and **S3 Table**) regardless of experimental genetic tractability (**Table 1**). These analyses can be used to identify drugs for repurposing or the best model system for testing a novel drug target. To do this, we sequentially removed each reaction from the reconstruction to identify which reactions are necessary for growth (*i.e.* production of biomass). These simulations are performed in an unconstrained model (*i.e.* all metabolites with a transporter can be imported, all enzymes can be used) to simulate the parasite's growth intracellularly in the nutrient-rich host cell. Dissimilarity of reaction essentiality for all *Toxoplasma*, *Plasmodium*, and *Cryptosporidium* reconstructions was calculated using the Euclidean distance (**Fig 7**).

**Table 2. Most frequently gapfilled reactions.** These reactions (in the BiGG namespace) were the most commonly added reactions as a result of all gapfilling steps.

| Reaction | Gapfilled N times? | Reaction Name |
|---|---|---|
| NADPPPS | 96 | NADP phosphatase |
| PYDXO | 90 | **Pyridoxal oxidase** |
| IMPtr | 86 | Transport of IMP |
| SO4HCOtex | 84 | Sulfate transport via bicarbonate countertransport |
| EX_lyslyslys_e | 81 | LysLysLys exchange |
| LYSLYSLYSr | 81 | Metabolism (Formation/Degradation) of LysLysLys |
| LYSLYSLYSt | 81 | LysLysLys transport |
| PSERT | 80 | Phosphoserine transaminase |
| PGCD | 75 | Phosphoglycerate dehydrogenase |
| GTHOXti | 74 | Glutathione transport |
| CYSLY3 | 65 | Cysteine lyase |
| NNDPR | 65 | Nicotinate-nucleotide diphosphorylase (carboxylating) |
| PDYXPT_c | 64 | **Pyridoxamine-pyruvic transaminase** |
| H2O2t | 63 | Hydrogen peroxide transport |
| PSP_L | 60 | Phosphoserine phosphatase (L-serine) |
| EX_ileargile_e | 59 | IleArgIle exchange |
| ILEARGILEr | 59 | Metabolism (Formation/Degradation) of IleArgIle |
| ILEARGILEt | 59 | IleArgIle transport |
| lipid2 | 59 | aggregation of all fatty acyl-CoAs |
| ASPTA6 | 58 | **L-alanine-alpha-keto acid aminotransferase** |
| GMPR | 56 | GMP reductase |
| GTHRDH_syn | 55 | Glutathione hydralase |
| GTHPe | 53 | Glutathione peroxidase |
| H2Ot | 51 | Water transport |
| HISD_c | 48 | Histidine degradation to glutamate |

Reaction essentiality is generally more similar for closely related organisms (*i.e.* within genera). However, some genera generate more similar predictions than others. Essentiality predictions were more similar when comparing *Plasmodium* genomes to one another than between *Toxoplasma*, despite all genomes being of the same species, or *Cryptosporidium* genomes. *Cryptosporidium* genomes generate predictions that are significantly less similar than *Plasmodium* genomes. Essentiality predictions in *T. gondii* are less similar to *Cryptosporidium* parasites than to *Plasmodium* (**Fig 7**). As *T. gondii* is a popular model system for other parasites, this result supports the use of *T. gondii* to test hypotheses about *Plasmodium* over *Cryptosporidium*. Moreover, we can identify organisms that are particularly unique within a genus. For example, *C. parvum* is a poor representative of *C. ubiquitum* whereas *C. muris* and *C. andersoni* are quite similar. Despite being distinct immunotypes, *T. gondii* VEG and GT1 are the most similar *Toxoplasma*. *P. vivax* Sal-1, an unculturable and clinically relevant *Plasmodium* species, is more similar to *P. knowlesi* H than the average two *Plasmodium* genomes, whereas *P. falciparum* 3D7 and *P. berghei* ANKA are among the most dissimilar *Plasmodium* genomes. Importantly, more complete models generate less similar predictions indicating differences in essentiality reflects functional differences, not merely incomplete genome annotation resulting in incomplete reconstructions (**S6 Fig**). These results highlight how ParaDIGM can be used to identify functional similarities and differences between parasites that directly inform experiments for developing and studying new drugs.

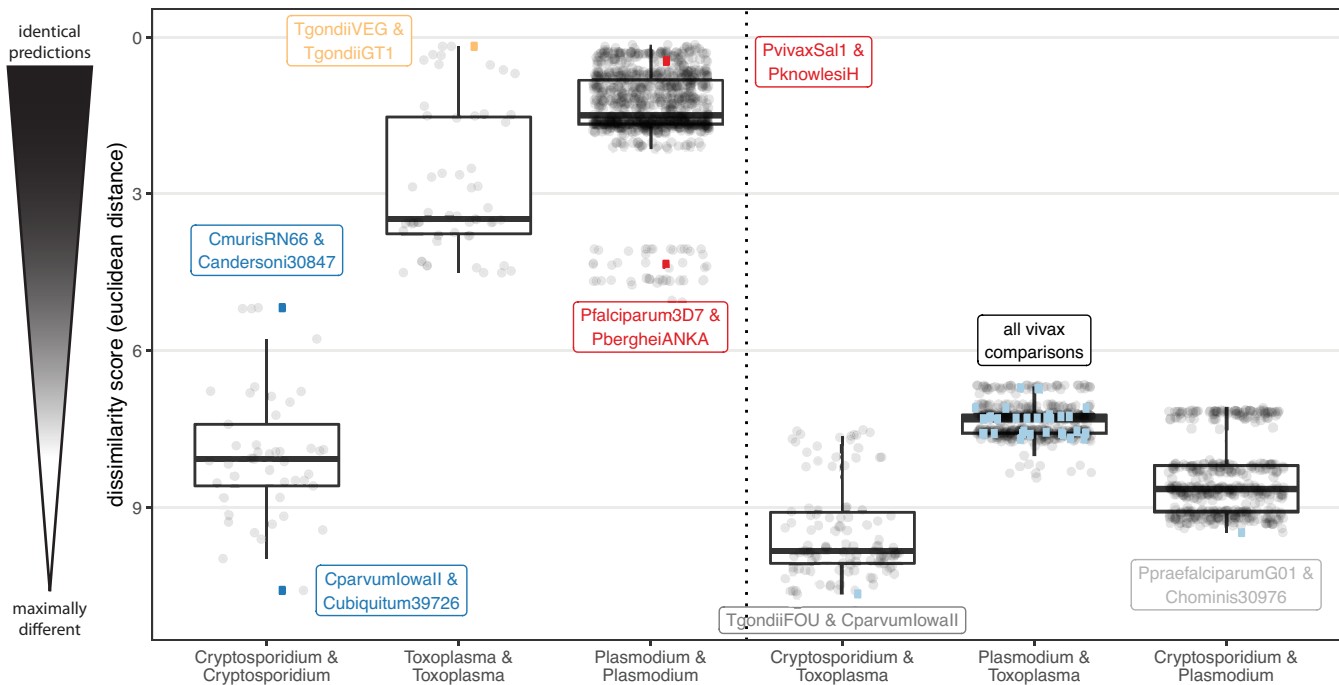

**Fig 7. Selecting experimental model systems using reaction essentiality.** Single reaction knockouts were performed on unconstrained models to identify the reactions that are essential for generating biomass. Dissimilarity scores were calculated from binary essentiality results using Euclidean distance (root sum-of-squares of differences). A low dissimilarity score of 0 indicates enzyme essentiality is identical between the two models; a high score indicates many differences. Each point represents a pairwise comparison with genera labels on the x-axis. Within genus comparisons are made on the left; across genus comparisons are made on the right. Several examples are highlighted with genome names. Genome-wide reaction essentiality is more similar between *Toxoplasma* and *Cryptosporidium* than *Toxoplasma* and *Plasmodium*. Mean dissimilarity score is significantly different (by two-sided student's t-test with multiple testing correction) between every labeled group.

## Discussion

Here, we presented a novel pipeline for generating metabolic network reconstructions from eukaryotic genomes and applied it to create 192 reconstructions for parasites, expanding the scope of parasite modeling. These reconstructions represent the first genome-scale metabolic network reconstructions for all but nine of these organisms, making ParaDIGM the broadest computational biochemical resource for eukaryotes to date. ParaDIGM uses reaction and metabolite nomenclature from the Biochemical, Genetic and Genomic knowledge base (BiGG, which includes both microbial and mammalian genome-scale metabolic network reconstructions) [56], facilitating future work involving host-pathogen interaction modeling. Gene nomenclature used in ParaDIGM is from the Eukaryotic Pathogens Database (EuPathDB) [54], consistent with the parasitology field standards and 'omics data collection. Reproducible data integration approaches are used to curate each reconstruction, making this the first fully automated reconstruction pipeline for eukaryotes; code and data are available in the **Data Availability** section for iterative improvements by ourselves and others.

ParaDIGM or individual reconstructions can be used for comparative analyses or applied to interrogate clinically- and biologically-relevant phenotypes. The adherence to community standards for metabolic modeling throughout ParaDIGM enables easier manual curation for users interested in studying a specific parasite in more detail. Together, this adherence to standards and the automated approach for integration of experimental data, will accelerate further curation of ParaDIGM itself as genome annotation improves, more experiments are

performed with individual parasites, and ParaDIGM users provide feedback on reconstruction usage and performance.

This eukaryote-specific reconstruction process (**Fig 2A**) generates comprehensive networks of comparable quality to manually curated parasite reconstructions (**Fig 2B-2D**). However, manual network curation and adding condition-specific constraints remain the gold standard approaches to maximize the accuracy of network predictions, especially for modeling stage-specific metabolism (i.e. [58]). Even so, our semi-automated curation approach enhances the genome-wide coverage of each reconstruction (**Fig 2C**) and generates models with comparable accuracy to previously published manually curated reconstructions (**Fig 2D**); maximum model accuracy is dependent on including compartmentalized reactions in the reconstruction process (**S7 Fig**).

To evaluate these networks, we compare *in silico* predictions to experimental results; all have imperfect accuracy regarding gene essentiality (**Fig 2D**), emphasizing how challenging it is to make a truly predictive model without integrating extensive stage-specific experimental data. High rates of false positives (when the model incorrectly identifies a gene as essential) are a product of the model building process; these reconstructions are built to summarize *all* metabolic capabilities of the organism, not the specific stage-dependent phenotype of an organism in the experimental system. Thus, constraining a reconstruction with *in vitro* expression data will reduce the false positive rate (*e.g.* [47,58]). We also compared our manually curated *P. falciparum* 3D7 reconstruction to our new semi-curated reconstruction for the same species. Differences in the two iterations fall into three groups: (1) non-specific genes that map to multiple reactions, (2) non-enzymatic genes (specifically, tRNAs), and (3) metabolic functions not yet encoded in the BiGG database for which reaction objects were created in our manual curation efforts. These differences can inform the first round of curation for semi-curated reconstructions.

However, this pipeline offers a few key limitations. First, simulation accuracy remains low (**Fig 2D**), largely because annotation pipelines may over-annotate function and (importantly) these models represent the metabolic capacity of many life stages, whereas experimental data is derived from a single timepoint. Thus, without constraining these models with stage-specific data, models will under predict essential genes and have poor accuracy. Furthermore, all reconstructions are limited by the data used for their construction; for example, we used the reactions already documented in the BiGG database as a universal set of reactions. Thus, reactions not contained in BiGG will not be included in ParaDIGM and only the reactions with specific cofactor utilization or directionality documented in BiGG will be included. Similarly, limited experimental data are available for the localization of specific enzymes and transporters and there has been limited successful experimental validation of computational predictions. Transporters in particular will influence pathway usage and a large proportion of transporters were added in the gapfilling process (**S8 Fig**). Lastly, the data incorporated in objective reactions and extracellular environment (i.e. media formulation) heavily influences which reactions are considered essential and which non-genetically supported reactions are added via gapfilling. Currently, these constraints within ParaDIGM are not experimentally-derived for each organism.

The metabolic capacity represented in ParaDIGM will be expanded and the accuracy of each reconstruction and associated simulations will be improved as (1) BiGG is further expanded, (2) more confidence is gained regarding protein localization, and (3) metabolomics analyses improve biomass and media formulation. Additionally, we used our orthology-based semi-curation approach for only *Plasmodium* models; however, this approach can be used for other organisms to propagate manual curation efforts (from our group and others, e.g. [48–50]) from one species to closely-related organisms as well. Finally, these reconstructions have

not been manually curated and require such attention for improved accuracy, especially in well-studied pathways and transporters and to represent stage-specific phenotypes.

Despite these limitations, the models within ParaDIGM perform similarly to manually curated models (**Fig 2D**) and so we highlight example use cases. First, we used ParaDIGM to better leverage model systems for drug development by identifying divergent or conserved metabolic pathways between select human pathogens. Network structures were quite unique with only 25.8% of all reactions in more than 50% of the reconstructions (**Fig 4**); network topology did however cluster by genus, and transport ability is associated with specific host environments (**Fig 5**). Despite these structural similarities, minor topological differences in networks (and unique functions, **Figs 3C and 4**) confer key metabolic strengths or weaknesses (**Figs 6 and 7**). We compare metabolic reaction (or enzyme) essentiality to identify the best *in vitro* system or non-primate infection model of disease for drug development (**Fig 7**). For example, enzyme essentiality is broadly more consistent between *Toxoplasma gondii* and *Cryptosporidium* parasites than between *T. gondii* and the malaria parasites. By leveraging network context (**Fig 6A and 6B**), we can impute fluxomic studies in all 192 parasites (**Figs 6D** and **S5**) to contextualize the variable results across species in relatively few *in vitro* fluxomics studies (**Fig 6C**) and to expand these observations to untested organisms and metabolites (**S5 Fig**).

Beyond our use cases of ParaDIGM, the pipeline and reconstructions presented here can be used broadly by the field. The study of microbial pathogens generated paradigm-shifting results in biology. The study of viruses revealed basic cellular machinery present nearly ubiquitously in eukaryotic cells, such as the discovery of alternative RNA splicing in adenovirus [66]. The study of bacteria has provided a nearly real-time observation of evolution, allowing researchers to perform hypothesis-driven evolutionary biology experiments in addition to observational research [67]. These microorganisms have shed light on cell biology and the history of life in impactful yet highly unanticipated ways; experimental challenges associated with parasites have slowed their utility in this regard. However, both the genetic 'dark matter' of eukaryotic parasites and known parasite-specific functions are abundant (**Fig 1**); thus, parasites too have the capacity to inform our understanding of life. The reconstructions in ParaDIGM can be used broadly to contextualize existing experimental data and generate novel hypotheses about eukaryotic parasite biochemistry as it relates to the rest of the tree of life.

ParaDIGM provides a framework for organizing and interpreting knowledge about eukaryotic parasites. The reconstruction pipeline designed for ParaDIGM implements and builds on field-accepted standards for genome-scale metabolic modeling and the latest genome annotations in the parasitology field; moreover, it is uniquely tailored to eukaryotic cells by recognizing the importance of compartmentalization and the design of the objective function. The pipeline can be implemented with other organisms and re-implemented iteratively to incorporate novel genome sequences, biochemical datasets, genome annotations, and reconstruction curation efforts. The genome-scale metabolic network reconstructions organized in ParaDIGM also can be used broadly by the scientific community, using the reconstructions as-is as biochemical and genetic knowledgebases or as draft reconstructions for further manual curation to maximize the utility and predictive accuracy of the models. These reconstructions can be used to generate targeted experimental hypotheses for exploring parasite phenotypes, ultimately improving the accessibility of modeling approaches, increasing the utility of parasites as model systems, and accelerating clinically-motivated research in parasitology.

## Supporting information

**S1 Table. Automated curation tasks.** All reconstructions were gapfilled to ensure the network could consume or produce all relevant metabolites outlined below. Data from multiple strains

of one species were aggregated. The first two columns describe each metabolite with a subsystem and name. The first two columns represent the genus and species for which literature evidence was compiled. Each {i,j} position in the matrix represents whether there is experimental evidence for a species' consumption or production of the metabolite. Blank cells indicate no literature evidence was found for that metabolite in that species.
(XLSX)

**S2 Table. Genetically-encoded transporters.** Transporters in each reconstruction prior to gapfilling. These transporters are annotated into each genome.
(CSV)

**S3 Table. Gapfilled transporters.** Transporters added to each reconstruction in the gapfiling process. These transporters are necessary to generate flux through the biomass reaction using parsimony-based gapfilling.
(CSV)

**S4 Table. Available essentiality datasets.** Experimental genome-wide essentiality datasets that are available in the literature. These data for *Toxoplasma* and *Plasmodium* were used to evaluate model performance and specifically simulated gene essentiality.
(CSV)

**S5 Table. Compartmentalization.** Subcellular compartments used for reconstructions in each genus.
(XLSX)

**S6 Table. Blocked reactions.** Each row represents a reconstruction (named in column 1). All following columns list out the BiGG identifiers for the blocked/unconnected reactions in that reconstruction. The *products* of blocked reactions are not used in any other reaction, whereas the *reactants* of unconnected reactions are not generated by any other reaction. Row headings are arbitrarily counting the number of reactions in each reaction in that category.
(XLS)

**S1 Fig. Comparison of ParaDIGM gene coverage to genes annotated on EuPathDB with GO terms relating to metabolism. (A): Schematic for gene count comparisons.** Genes found on EuPathDB with a GO term related to 'metabolic processes', genes incorporated into *de novo* reconstructions (N = 192 reconstructions), and genes in both of these categories are described. **(B): Number of genes per category.** Boxplots represent the total number of genes per category across all reconstructions. **(C): Number of genes EuPathDB associated with three example pathways.** These genes represent the intersection of EuPathDB genes and reconstruction genes for three select pathways. For all panels, the box extends from the lower to upper quartile values of the data; the center line marks the median and whiskers shows the range of the data. Outliers are not shown. **(D): Percent of genes on EuPathDB associated that are represented in reconstructions.** Again, these genes represent the intersection of EuPathDB genes and reconstruction genes for three select pathways. For each example pathway in Panel C, the percent of total genes from EuPathDB that are represented in the *de novo* reconstructions are shown.
(EPS)

**S2 Fig. Reaction content overlap between Plasmodium reconstructions in ParaDIGM and a manually-curated reconstruction, iPfal22.** Semi-curation improves reaction content overlap between *Plasmodium* reconstructions in ParaDIGM and a well-curated reconstruction. Venn diagram of reaction content between three *Plasmodium* species or strains (*falciparum*

3D7 in **A**, *falciparum* Dd2 in **B**, and *berghei* ANKA in **C**) for the draft reconstruction including only genes identified by Diamond and the semi-curated reconstruction (see *Additional Information*: *Online Methods*), and compared to iPfal22.
(EPS)

**S3 Fig. Benchmarking of gapfilling approach on Plasmodium reconstructions.** Only *Plasmodium* reconstructions were assessed here because we can compare the resultant reconstructions to a manually-curated reconstruction, iPfal22. **(A):** Number of reactions added at each step of the reconstruction pipeline per reconstruction. For each boxplot, the box extends from the lower to upper quartile values of the data; the center line marks the median and whiskers shows the range of the data. Outliers are not shown. **(B-D):** Venn diagrams highlighting the shared reaction content of three *Plasmodium* reconstructions with and without gapfilling. Gapfilling increases the coverage of functions contained in the well-curated reconstruction, iPfal22.
(EPS)

**S4 Fig. Further characterization of ParaDIGM reconstructions. (A): Unique reactions by reconstruction** 39 reconstructions contain at least one unique metabolic reaction, or a reaction not found in any of the other 191 models. Reconstructions are colored by EuPathDB grouping, like in panel A, and the bar represents the number of unique reactions in that reconstruction. **(B-C): Unique reactions are well connected.** Percent (B) and total number (C) of all unique reactions per reconstruction that are blocked, unconnected, or both blocked and unconnected. Blocked reactions are defined as those whose products are not utilized by any other reactions (including transport and exchange reactions), whereas unconnected reactions are those whose reactants are not produced by other reactions. For all panel B, the box extends from the lower to upper quartile values of the data; the center line marks the median and whiskers shows the range of the data. Outliers are shown as points. For panel C, each column or bar represents an individual reconstruction. Most unique reactions are connected and unblocked. **(D-E): Gapfilled model size remains correlated with genome size.** Following gapfilling, the relationship between genome size (as measured by open reading frames or genes with metabolic GO terms, **D** and **E** respectively) and model size remains; however, the correlation is weak due to an increase in the number of reactions for reconstructions built from medium-sized genomes. **(F-G): Larger genomes have more unique reactions before (F) and after (G) gapfilling.** Genome size is measured by open reading frames. For panels D-G, line is fit to a linear regression with R2 noted (p-value < 0.001); the standard error is not shown. Points in **D-G** are color coded by database.
(EPS)

**S5 Fig. Complete in silico metabolic capacity.** Inferred metabolic capacity of each organism (rows) for metabolites (columns) for every reconstruction and metabolite in ParaDIGM (5,141 metabolites by 192 models). See *Fig 6B* for definitions. Note the sheer volume of data acquired from ParaDIGM.
(EPS)

**S6 Fig. Evaluation of prediction dissimilarity and reconstruction coverage. (A-C):** Distribution of reconstruction completeness scores for *Cryptosporidium*, *Plasmodium*, and *Toxoplasma* reconstructions. Completeness scores were calculated by identifying the ratio of metabolic functions (reactions) in a single reconstruction compared to the sum of metabolic functions covered by all reconstructions from the respective genus. **(D):** Prediction dissimilarity is correlated to model completeness. For each pair of models, reaction essentiality predictions were compared to generate a dissimilarity score (**Fig 7A**). Identical predictions have a

dissimilarity score of 0 whereas a high dissimilarity score indicates divergent predictions.
(EPS)

**S7 Fig. Compartmentalization improves prediction accuracy and coverage of the genome.**
**(A)**: Sensitivity and specificity of *in silico* gene essentiality predictions when compared to
experimental data. Reconstructions generated using our pipeline with and without compart-
mentalization are connected with a line; other points represent published models for reference.
Our compartmentalization approach improves the sensitivity and or specificity of gene essen-
tiality predictions for *P. berghei* ANKA, (P. falciparum) 3D7, and *T. gondii* GT1. **(B)**: Venn dia-
gram of gene content for the ParaDIGM P. falciparum 3D7 reconstruction (with and without
compartmentalization) and our manually curated P. falciparum 3D7 reconstruction, iPfal22.
Incorporating compartmentalization improves genome coverage by adding 12 genes also
found in iPfal22 and 17 genes not found in iPfal22.
(EPS)

**S8 Fig. Only half of transporters are genetically supported. (A):** Percent of all transporters
with gene annotations. **(B):** Percent of intracellular transporters with gene annotations. Red
dotted line indicates mean.
(EPS)

**S9 Fig. Distribution of blocked and unconnected reactions.** The distribution of poorly con-
nected reactions (A: blocked, B: unconnected, C: both) was similar before and after gapfilling.
(TIFF)

## Acknowledgments

The authors would like to acknowledge the helpful discussion and feedback from members of
the Petri, Mann, Guler, and Papin labs, as well as Drs. Alison Criss, Norbert Leitinger, Herve
Agaisse, and Young Hahn. The authors would also like to thank the University of Virginia
ARCS staff for support regarding UVA's High Performance Computing cluster, especially Kar-
sten Siller and Katherine Holcomb. Lastly, the authors would like to thank the EuPathDB,
BiGG, and CobraPy communities for providing essential tools (software and database infra-
structure) and inspiration.

## Additional information

### Online methods

**Code dependencies**

R [68] and R packages tidyverse, ggpubr, ggdendro, seqinr, Biostrings, msa, reshape2,
UpSetR, cluster, ade4, RColorBrewer, readxl, dplyr, and ggdendro were used for analysis or
visualization [69–80]. Python 3.6.4, pandas [81], and CobraPy 0.14.1 [55], as well as code to
implement Diamond-based annotation scoring from CarveMe [52], were used for the recon-
struction and modeling. Memote [62] was used to evaluate all reconstructions.

**Genomic analyses**

Sequences were obtained from EuPathDB release 44 [54]. EuPathDB curates and compiles
genome annotation for all genomes hosted by the database. We used open reading frames
identified on EuPathDB and annotated the sequences with Diamond, described below.
EuPathDB's OrthoMCL was used for mapping orthology between *Plasmodium* species. In
brief, orthology was mapped within each EuPathDB database by the 'map by orthology' tool
from the genome of each organism with a curated reconstruction to all other genomes within
that database. The search protocol was 'new search > genes > taxonomy > organism {pick} >

transform by orthology'. We mapped each organism's amino acid sequences using Diamond annotation [82] against proteins referenced in the BiGG databases [56]. Diamond is a similar approach to BLAST, with sensitive and fast performance on protein annotations [82].

**Model generation**

We generated draft reconstructions by first annotating each organism's amino acid sequences, obtained from EuPathDB [54], using Diamond annotation [82] against enzymes and transporters referenced in the BiGG databases [56]. We next mapped all functional annotations to reactions contained in the BiGG database [56] inspired by the approach conducted with the reconstruction pipeline CarveMe [52]. Next, for all reconstructions, we added any KEGG reaction-associated genes from EuPathDB if the KEGG identifier could be converted into a BiGG identifier, using KEGG-BiGG mappings found at http://bigg.ucsd.edu/static/namespace/bigg_models_metabolites.txt. Importantly, EuPathDB annotations used for this KEGG-BiGG mapping are derived from a mix of variable automated and manual approaches and include user contributions. Methods are included in the analytic code hosted on Github, at https://github.com/maureencarey/paradigm.

Unlike the CarveMe approach [52], we included all high-scoring reactions. CarveMe maximizes the number of high-scoring hits while building a functional network. Our conservative approach generates broadly inclusive but incomplete reconstructions (*i.e.* that are not able to produce biomass until gapfilled). This approach added redundant reactions from multiple different compartments (*e.g.* peroxisome, mitochondria, and cytosol) so all reaction versions other than the cytosolic version were pruned unless localized to a relevant compartment (**S5 Table**); for genera not included in **S5 Table**, only the cytosol and extracellular space were used to avoid inclusion of erroneous compartments. For example, if a *Plasmodium* reconstruction contained a reaction in the cytosol, mitochondria, and chloroplast, only the cytoplasmic and mitochondrial reaction versions would be kept. Following this step, a large percentage of each reconstruction's reactions remained in unsupported compartments because there was no analogous cytosolic (or extracellular) reaction (**Fig 2B**). Next, reactions only found in an unsupported compartment were moved to the extracellular space or cytosol; specifically, periplasmic metabolites were moved to the extracellular space and all internal subcompartment metabolites were moved to the cytosol. However, this step removed all reactions that summarized a transport reaction from the extracellular space to periplasm or from the cytosol to an unsupported organelle. The extracellular compartment corresponds to the parasitophorous vacuole space contained within the host cell for intracellular parasites (*i.e. Plasmodium*, *Toxoplasma*, *Cryptosporidium*) and the host serum for extracellular parasites (*i.e. Trypanosoma*).

**Manual curation**

We performed brief manual curation from literature sources, building on our curation conducted in [47,57]. Manual curation included updating reaction identifier and notes fields for consistency to CobraPY standards, updating metabolite identifiers for consistency to BiGG standards, replacing old PlasmoDB gene identifiers with updated identifiers, mass balancing seven reactions (with phosphate or hydrogen), removing duplicate reactions and metabolites (12 modifications), pruning unused metabolites, ensuring all metabolites have associated compartments, adding annotations to reactions and metabolites if missing (i.e. KEGG identifiers, EC codes, InChI strings), and adding SBO terms to all objects. Our previous asexual blood-stage *Plasmodium falciparum* 3D7 reconstruction ([57] adapted from [47]) was manually curated generating *iPfal22*. See **Data Availability** for code documenting all for modifications and implementation of manual curation.

Additional manual curation was performed on lipid metabolism of the asexual blood-stage *P. falciparum* using the lipidomics study presented in [83] adding over 700 reactions, 400 metabolites, and 18 genes. This curation removed aggregate reactions representing lipid

metabolism and replaced them with individual reactions for individual lipid species, as supported by the lipidomics study. This model is available, but was not used for the analyses presented here as the metabolic demands for lipids in our biomass reaction are also aggregated. Inclusion of these reaction is appropriate for understanding lipid metabolism but would create random distributions of flux through the individual reactions that may distract from meaningful changes in flux distributions. All code for this curation is available at https://github.com/gulermalaria/iPfal17_curation.

**Automated orthology-driven curation**

We developed a novel automated curation approach using orthologous transformation, similar to the approach taken by [48]. Our approach leverages the curation conducted in one organism for closely-related organisms and applied it to all draft *Plasmodium* reconstructions using *iPfal22* (**Fig 2A**). We first mapped orthology of *P. falicparum* to each other *Plasmodium* species to build an orthology thesaurus (**Fig 2A**). We then added genes and associated reactions from *iPfal22* if there was an orthologous gene in the target species' reconstruction and the reaction was not already present (**Fig 2A**) resulting in a significant increase in the number of genes and reactions in each reconstruction (**Fig 2C**). Notably, this approach facilitates the compartmentalization of these reconstructions, a function most automated pipelines except RAVEN [45] and merlin [46] fail to include. This step is particularly important for parasite-specific compartments such as the apicoplast, which is not included in any database.

**Gapfilling (part 1)—Automated data-driven curation**

Gapfilling is an analytic process used to bridge or complete genetically-supported metabolic pathways to permit the network to fulfill metabolic functions, and was used to generate functional models. To increase the scope of a reconstruction (*i.e.* to add reactions), we performed gapfilling to fill in gaps in pathways to ensure that the reconstruction can complete a particular task. This optimization process adds reactions to allow the reconstruction to carry flux under given constraints.

Thus, automated curation of all reconstructions was performed by gapfilling for metabolites measured to be consumed in fluxomic or select media formulation studies. Detailed analysis is provided in our code. In brief, following an extensive literature review, we compiled data providing evidence for consumption or production of select metabolites (**S1 Table**). Metabolites were defined as consumed by the parasite if: (1) the metabolite was radiolabeled, added to media, incorporated into the parasite or converted by the parasite, and this was not seen to the same degree in uninfected host cells; (2) the metabolite rescued inhibitor treatment of a metabolically upstream parasite enzyme; or (3) the metabolite is an essential media component for parasite culture. Metabolites were defined to be produced by the parasite if the metabolite was radiolabeled following growth in a media containing a radiolabeled precursor metabolite, and this was not seen to the same degree in uninfected host cells.

To gapfill for these tasks, import or excretion of these metabolites were added to the reconstruction. Next, the model objective was changed to an internal demand reaction for the metabolite or excretion reactions, respectively, and was gapfilled sequentially; this ensures import or synthesis of each of these measured metabolites. We then used a parsimonious flux balance analysis (pFBA)-based approach as originally used in [60] and further developed in [61] to add the minimal number of reactions required to carry flux through the internal demand or excretion reaction. Code is linked in the **Data Availability** section. In essence, this pFBA approach minimizes the flux through genetically unsupported reactions from a biochemical database such that the network can carry flux through the objective reaction (*i.e.* metabolite production or consumption or biomass synthesis). Any reaction from the database that carries flux during this problem was added to the network.

**Gapfilling (part 2)—Biomass as the objective function**

After compartmentalization, manual curation, and gapfilling for individual metabolites, we used pFBA-based gapfilling to ensure each network was capable of generating biomass. We use two classes of biomass functions here to robustly evaluate model performance, specifically a genus-specific curated biomass reaction (for *Plasmodium* reconstructions) and a parasite-specific generic biomass reaction. The genus-level curated biomass reaction was developed and curated for our manually curated *Plasmodium falciparum* model; see [47] for justification of this biomass reaction. Our generic biomass contains the consensus set of metabolites from several manually curated reconstructions for *Plasmodium falicparum*, *Leishmania major*, and *Cryptosporidium hominis*. The stoichometry for this generic biomass is from the *iPfal22* biomass reaction [47]. Unfortunately, variability in reconstruction namespace (*i.e.* the database used for metabolite and reaction naming conventions) make it difficult to reconcile data compiled for some parasite reconstructions, such as the *Toxoplasma* and *Plasmodium* reconstructions *ToxoNet1* in the KEGG namespace and *iPfal22* in the BiGG namespace, as there are not always one-to-one mappings of variables across databases. This generic biomass was used to capture the most conservatively defined required *parasite* biosynthetic capacity. All reconstructions were gapfilled to ensure biomass could be synthesized via the generic parasite biomass reaction; all *Plasmodium* reconstructions were also gapfilled to ensure biomass could be synthesized via the genus-specific biomass reaction. For all biomass gapfilling steps and simulations, all exchange reactions were kept open to simulate a nutrient rich extraparasitic environment. Blocked and unconnected reactions were kept in the reconstructions (**S6 Table**). The distribution of blocked and unconnected reactions were similar in *de novo* and gapfilled reconstructions (**S9 Fig**).

**Model performance evaluation**

Network accuracy was evaluated against gene essentiality data (**S4 Table**) if available. Gene essentiality was simulated by performing single gene deletion studies in our models. All simulations were performed in a model state without additional experimental constraints; specifically, all exchange reactions were permitted to carry flux reversibly, simulating a nutrient rich environment such as intracellular growth. Of note, these 'unconstrained' models do not necessarily represent the *in vitro* or *in vivo* environment in which all experiments were conducted, merely the metabolic capacity an organism encodes. If this extracellular environment was constrained to represent the specific host environment, we would see further separation of model predictions.

Gene deletions were simulated by removing the gene of interest from the model using CobraPy's 'single_gene_deletion' function. This change results in the inhibition of flux through all reactions that require that gene to function. If the model could not produce biomass with these constraints, the gene was deemed essential. Specifically, we defined an essential gene as a knockout that resulted in 10% or less of the maximal biomass (measured in mmol/(gDW*hr)). Knockout accuracy was defined as the sum of true positives (refractory to knockout or lethal genes) and true negatives (nonlethal genes) divided by the total number of predictions. As targeted metabolomics data were used for model generation (by gapfilling), we excluded these data from the evaluation data.

Models were tested for thermodynamically-infeasible loops and energy-generating cycles; the approach outlined in [84] was used with minor modifications for eukaryotic cells.

**Memote evaluation**

Models were evaluated using Memote [62]; example outputs are presented on https://github.com/maureencarey/paradigm/tree/master/memote_reports. Additionally, Memote was used to quality control the reconstructions throughout the development of the ParaDIGM pipeline to improve standard compliance (especially annotation coverage) and biological relevance (*e.g.* network connectivity and topology).

### Reconstruction pipeline for eukaryotes

Our pipeline itself (**Fig 2A**) is uniquely tailored to eukaryotic cells by recognizing the importance of compartmentalization and the design of the objective function. Compartmentalization biases predictions [47]; both our *de novo* reconstruction and orthology-driven approaches addresses this important step. Compartmentalization was incorporated into our *de novo* reconstruction pipeline and implemented for several genera (**S5 Table**). We adjusted the localization of reactions if inappropriately compartmentalized and added unique parasite-specific compartments (*e.g.* the apicoplast in *Plasmodium*). This addition was done by adding genetically supported reactions were to all feasible compartments. If a gene-encoded enzyme corresponds to both mitochondrial and cytoplasmic reactions, both versions will be included: adding network redundancy that may not be biologically accurate. Alternatively, if an enzyme maps to a chloroplast reaction, our approach moved the reaction to the cytosol. It is plausible that chloroplast reactions like this example are not catalyzed by the parasite. These modifications are encoded in our analytic pipeline for future reference (see code, **Data Availability**). Additionally, we used a curated model to inform the compartmentalization of each semicurated model (**Fig 2A**); genes associated with compartmentalized reactions were mapped via orthology, assuming orthologous genes has comparable localization across species.

Similarly, the objective function (often representing biomass synthesis [85]) influences network simulations [86] and the assumptions used in formulating a biomass reaction for prokaryotes may not apply to eukaryotes [87]. For example, in the first genome-scale metabolic model of any *Cryptosporidium* species, *C. hominis* [88], 30 reactions involved in lipid synthesis were unsupported by genetic evidence and manually added to the network. The selection of lipid species as biomass precursors may have impacted the addition of these unsuported reactions and resultant simulations with the model; for example, the 30 gapfilled reactions might not be added if alternative or fewer lipid species were included in the biomass reaction.

Thus, we use a consensus biomass reaction derived from multiple curated parasite reconstructions, as well as a genus-specific biomass for *Plasmodium* reconstructions. These objective functions influence reactions added via gapfilling, adding uncertainty in network structure [60,61]; thus, we emphasize which reactions were gapfilled frequently (*i.e.* for many species within a genera or ParaDIGM broadly) to increase our confidence in these predicted functions. Targeted investigation of these reactions (such as pyridoxal oxidase and shikimate dehydrogenase) will increase our confidence in all of our parasite network reconstructions.

Furthermore, initial *de novo* reconstructions were supplemented with curated information from the EuPathDB databases. All genes associated with KEGG reaction identifiers were collected from EuPathDB. These KEGG identifiers were then converted to the BiGG namespace, whenever possible, and added these reactions to each genome reconstruction. If the reaction was already present but the gene was not, the gene was added to the reaction's gene-protein-reaction rule (using an 'OR' relationship if another gene already was included). This process improved the scope of the network reconstruction for each species/genome and leverages one of the best features of the EuPathDB family of databases: user contributions and literature support. This builds on the bioinformatic approach (i.e. Diamond annotation) to build *de novo* reconstructions by integrating curated gene annotations that have been documented on EuPathDB.

Even with this addition, it is important to note that not every gene with a metabolic annotation on EuPathDB will be included in the reconstructions for three main reasons: (1) some metabolic processes are not included in genome-scale metabolic network reconstructions {group 1}, (2) some functions are insufficiently characterized to include as a reaction {group 2}, and (3) unfortunately not all KEGG IDs can be converted to the BiGG namespace {group

3}. For example, 1032 genes are on EuPathDB for *T. gondii* ME49, associated with a metabolic GO term, and not found via our Diamond annotation of metabolic genes. This list of genes includes TGME49_201130, TGME49_321660, and TGME49_326200. TGME49_201130 is annotated as a rhoptry kinase family protein ROP33a with GO terms including 'ATP binding' and 'protein kinase activity'; rhoptry biology is not traditionally the kind of function that would be included in genome-scale metabolic network reconstructions because it is not a bio-synthesis or degradation. Thus, this gene is associated with group 1 above. On the other hand, it may be appropriate to include the function associated with TGME49_321660 in a recon-struction as its associated GO terms include 'transferase activity' and 'transferring glycosyl groups'. However, the annotation for TGME49_321660 is not detailed enough to include in a reconstruction (i.e. there is no EC number or KEGG ID), so this gene belongs in group 2. Lastly, TGME49_326200 encodes a type I fatty acid synthase (EC 1.6.5.5) is associated with the KEGG ID (R02364) but no BiGG ID {group 3}. The groups 2 and 3 (2: incompletely annotated genes and 3: those associated with only KEGG IDs) are excellent starting points for manual curation.

In addition, to curate these reconstructions, we recommend readers focus on false positives, or enzymic functions that were identified but do not occur *in vivo* or *in vitro*. For example, when the model incorrectly predicts a function or gene is not essential, there is inaccurate redundancy upstream of the prediction; thus, a curator can use these observations to identify and remove false positive gene annotations.

## Author Contributions

**Conceptualization:** Maureen A. Carey, Gregory L. Medlock, Jason A. Papin.

**Data curation:** Maureen A. Carey, Michał Stolarczyk.

**Formal analysis:** Maureen A. Carey.

**Funding acquisition:** Maureen A. Carey, Gregory L. Medlock, William A. Petri, Jr., Jennifer L. Guler, Jason A. Papin.

**Investigation:** Maureen A. Carey, Gregory L. Medlock.

**Methodology:** Maureen A. Carey.

**Project administration:** William A. Petri, Jr., Jason A. Papin.

**Software:** Maureen A. Carey, Gregory L. Medlock.

**Supervision:** Jennifer L. Guler, Jason A. Papin.

**Validation:** Maureen A. Carey.

**Visualization:** Maureen A. Carey.

**Writing – original draft:** Maureen A. Carey.

**Writing – review & editing:** Maureen A. Carey, Gregory L. Medlock, Michał Stolarczyk, William A. Petri, Jr., Jennifer L. Guler, Jason A. Papin.

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
