## [Decision Letter · Decision Letter 0]

20 Oct 2021

Dear Dr. Carey,

Thank you very much for submitting your manuscript "Comparative analyses of parasites with a comprehensive database of genome-scale metabolic models" for consideration at PLOS Computational Biology. As with all papers reviewed by the journal, your manuscript was reviewed by members of the editorial board and by several independent reviewers. The reviewers appreciated the attention to an important topic. Based on the reviews, we are likely to accept this manuscript for publication, providing that you modify the manuscript according to the review recommendations.

Sincerely,

Pedro Mendes, PhD

Associate Editor

PLOS Computational Biology

Kiran Patil

Deputy Editor

PLOS Computational Biology

[LINK]

Reviewer's Responses to Questions

**Comments to the Authors:**

Reviewer #1: I congratulate the authors for creating a very important knowledge-base to the parasite community. The paper is very well written, the methods used are solid and the results reported are of extreme importance for the scientific community. I especially appreciate that the authors propose that the method will be used iteratively and models can be improved as more data arise.

As I mention in the revision below tha I believe that more and more the scientific community should be careful when only propagating data from closely related species. And the authors indeed show in their results that "Phylogeny is not the sole predictor of model similarity". This idea can also be extended to the genomic prediction, annotation, metabolic models reconstruction, and so on and we should keep in mind that much information might be lost when we try to tailor-made the models based mostly on related species.

This is also why I believe that creating this extremely rich and diverse knowledge-base becomes of utter importance to future advances in the area.

Comments and questions:

1. You describe that reconstructions were built based on the annotated genomes, and then proteins were re-annotated using Diamond against the BiGG database.

However, I would like to know and you could also discuss this within the paper about the genomic annotation present within EuPathDB, as I don't imagine that all annotations are done in a homogeneous way across distinct databases.

And if this is correct, how do you handle these annotations coming from different sources? Could you discuss in terms of how robust is your pipeline taking into account this kind of differences?

It would be interesting to also include for each genome within your dataset the proportion of uncharacterized protein coding genes or genes having DUFs (domains of unknown function) and whether these correlate with the proportion of reactions that are gap-filled in the final step.

2. In terms of compartmentalization, which was the proportion of transport reactions included in the models in order to allow for correct fluxes of reactions within compartments and how many of these are orphan reactions? Is this one the reasons you did not include compartments for all species or were there other reasons for this choice?

3. In page 4 (lines 7 to 11) and in Supp. Fig 1: I am not sure if I understood Suppl Figure 1C and D correctly. Are these the categories of genes that are only in EuPathDB (C) or these are all genes in EuPathDB? Also, regarding these genes that were not included in the reconstructions and by referring also to the methods described on page 17 lines 23-40: what is the distribution of genes belonging to each of the 3 groups you mention? And do these percentages change depending on the database used (for example TritrypDB versus ToxoDB)?

4. You mention that you perform a brief manual curation of models. I would like to know time-wise how long is brief in this case for comparison reasons to fully curation that can take up much time. This is a minor comment, but I think it would be important to state in the text whether the authors are planning to perform this type of curation with other species. This could be discussed as a perspective as it would be of great interest to the broader parasite community to have models that were constructed similarly in order to be comparable within different taxa.

5. Can you describe a little more the reactions that were present in the manual curation that were not added to the final semi-curated Plasmodium models? Why do you think these are not included and is it possible to propose a way to improve this step of semi-curation in the future in order to arrive at models closer to curated ones?

6. In Figure 3: You could use the colors of databases provided in 3A and 3C on the individual points for Figure 3B.

Also, it seems that not only number of reactions but also number of unique reactions are related to genome size, correct? It would be interesting to see the same trend as showed in Figure 3B but also taking into account unique reactions, especially since your barplots in 3C do not account for per genome variability. I would also highlight the genomes that actually deviate from this linear regression curve in order to pinpoint whether these differences arise from biological traits or if they are due to technical differences.

7. Still in Figure 3 (and the text): Could you explain why you don't find basic pathways such as glycolysis, TCA cycle, Pentose Phosphate, etc within your core reactions? It would be interesting to know in which models these reactions are not taking place and not only knowing they take place in ~90% of the models. Can you also explain if this is merely a technical issue of if this is in accordance with the biology of these species lacking central metabolism enzymes?

8. In page 5, line 33 you mention that even small reconstructions contain unique reactions prior to gapfilling. I am also wonder if these numbers change a lot after the gapfilling process.

9. I am not sure if there is a particular reason the authors do not discuss or compare also previous databases of metabolic networks such as Metacyc. I would at least advise you to explain if you perform better and if possible add an explanation as to why or state if these are not comparable instances of models from the same species. Also this is of interest since as you mention in page 11 line 3, reactions not included in BiGG database are not within your set of universal reactions.

10. Also, it was not clear to me whether the authors will regularly maintain and update the generated databases as new data arise or if this was more of a proposition that others can use the code in order to generate new models.

Finally, in my opinion, an extremely interesting finding is also buried within the "Niche-specific metabolic function" paragraph and could be brought much more into play either in discussion or conclusion if the authors are confident enough, which is:

"Phylogeny is not the sole predictor of model similarity".

And I say this at the end because I believe that your findings support the idea that we should not just propagate ortholog reactions from the most similar species available because much of the information might be lost by doing this.

Minor comments:

0. I could not find table 1 within the material provided.

1. In Figure 2C, I would label the "unlabeled dots" or at least provide a subtitle in terms of which species each is referring to.

2. Supplementary tables could have more information in tabs, column names and rows within the tables to improve readability. Also In Suppl. Table 4 I don't understand if the category "both" should be a sum of the other two (blocked/unconnected). If this is the case, I don't think the numbers add up.

3. In Suppl Figures 2 and 3, the blue color circle on top of a purple color does give a good contrast to interpretation.

4. In Suppl Figure 3A: Do you see different trends from distinct databases regarding the number of reactions added at each step or is this very similar to all groups?

4. Page 5 line 31: I believe the sentence ends abruptly and is missing some words.

5. In Figure 3E you could indicate from which database the species having the 7 metabolic unique reactions are coming from for easier readability.

6. In general I don't see the point of including "data not shown", so if it's not relevant to your paper, I would just exclude it, otherwise please support the information necessary for this statement.

7. Page 7, line 4: "Most important variables" are variables here related to "reactions"?

8. In page 10: you created the reconstructions for all "but nine" organisms. Which are these nine organisms? And why were they excluded? I found the explanation for T. gondii RH, but could not find other examples of excluded genomes.

Reviewer #2: This is a timely and important database. The paper is well-written. I have a few comments below, one of them more pressing, on the importance of transporter annotations.

The Introduction is very good at presenting the problem and need for this study, but it lacks a bit in presenting the state of the art for similar databases of GSMs-genome-scale models-and large-scale batch reconstruction efforts (e.g. ModelSEED, CarveME, BiGG, etc), as well as for large-scale comparative works with GSMs (mostly done in the aforementioned automated reconstruction tools, or evaluation tools e.g. MEMOTE). Comparisons of curated GSMs are very few, and use few models, (e.g. Xavier et al. Plos Comp Bio 2018) which highlights the need for databases as this one. I would advise a new paragraph before the last of the Introduction to do this - the authors already do mention much of this literature throughout the rest of the paper. In theory, the community as a whole would benefit from one, centralized source of good GSM data, (e.g. as done for genome data with NCBI), yet new ones continue to appear. Why is ParaDIGM important in this scenario, and how does it not increase division? This can be returned to in the Discussion as well. The usage of MEMOTE and BiGG nomenclature by the authors is commendable, and shows concern for community standards.

Fig. 1A is more of a table than a figure. I think for the community it would be more helpful as a table, annotated with the references that convey the information given. To keep it as a figure though, it would really help to ditch the current grey color that does not convey information, and perhaps color code "yes" and "no"; it would also really help to sort the table, perhaps by the last column, perhaps split it by disease (which could come in a first column, non-repeated).

How was KEGG information mapped to BiGG? Where is this mapping available for review? Please detail in the methods.

The most sensitive point: could you please expand on the process of annotation of transporters in the methods and discussion, perhaps consider a small analysis on how many were annotated per genome, and how many added through gap-filling? These have a huge influence in GSMs, but in the case of parasites it becomes extreme (more so if one considers transport between compartments, important in the case of eukaryotes as the authors highlight). So as the authors recognize particularly with the correlation of some results with the environment of each species (Fig.4) transport will have a major impact in this work. KEGG does not have transporter annotations in the form of reactions, so they must come from Diamond-BiGG + Gapfilling, but how much of each? Being very explicit about this point will increase the value of ParaDIGM substantially.

In general the Figures are too crowded, with much text and schematics - things that would be better as boxes, tables or supplementary material. Some colors are undefined in the figures themselves; some figures are just crowded with gridlines etc., e.g. fig 5C the grey squares are confusing as the grey color is part of the key. This also occurs in Supplementary Figures, e.g. Supp Fig 5 - there's an excess of color and data and it's hard to discern what one can infer from this figure. Please review the figures in general for readability, consistency, and to remove excess content.

Lines 19-26 page 11 - the authors mean 4D instead of 4C, twice, I believe

Extensive work required on reference formatting; some names in format "Last, First" others in "First Last", urls heterogeneous, etc.

Reviewer #3: 1)“Reaction compartmentalization was adjusted to maintain each gene-protein-reaction mapping but only with the subcellular compartments relevant for each organism. A large proportion of parasite gene-reaction pairs would otherwise be misassigned or removed from the network due to assignment to an incorrect compartment, due to lack of orthologous and compartmentalized reactions in biochemical databases; our pipeline reassigns these reactions to the cytosol (Figure 2B).”

-This should be explained more clearly as far as the” adjustments” that are happening. To be sure, does Figure 2B show only the percentage of reactions assigned to the cytosol? When I look at the legend below, I see that adjustment from the periplasm to extracellular is mentioned, not just cytosol re-assignments.

2)“B): Considering compartmentalization. Our approach moves a large proportion of the reconstruction’s reactions from compartments in a biochemical database to biologically-relevant compartments (e.g. periplasm to extracellular)."

- “biologically-relevant compartments (e.g. periplasm to extracellular).”Is the periplasm not a biological-relevant compartment? Although I understand why to assign periplasm to extracellular as an abstraction, I don’t understand how the point being made about compartments in a database vs biologically-relevant compartments in the context of the example provided.

-It is excellent that the authors are accounting for compartmentalization as they mentioned “, our de novo reconstruction approach accounts for compartmentalization, unlike previous metabolic network reconstruction pipelines.” I am unsure that the statement is correct. I believe that at least the metabolic reconstruction pipeline “merlin” accounts for compartmentalization as well (unsure about other tools that might have been released more recently).

-A little more clarification/changes in the paragraph text and legend would be appreciated to clarify this.

-Also, I only see the “pros” of making this compartment adjustment; as the authors mention, “large proportion of parasite gene-reaction pairs would otherwise be misassigned or removed”. What about the cons of doing this re-adjustment of reaction compartments and keeping those reactions?

3)"(D): Prediction accuracy. Semi-curated reconstructions (filled squares and triangles)"

-In the figure legend, filled squares are shown as "uncurated".

4)“semi-curated reconstructions 8 are larger in scope than de novo reconstructions (Figure 2C) and generate predictions with comparable or improved accuracy (Figure 2D).”

-What does the increase in scope provide? What are we gaining from all those "extra" reactions when we only get "comparable accuracy" in the models? I looked at the supplementary material data to see if this increase in the number of reactions also meant an increase in the number of blocked reactions in the model but could not find out.

5)"The relationship between genome size and model size is weakened following gapfilling (Supplemental Figure 4D)"

-Can you explain why this happens? I guess that biomass requirements/biomass equation will be the same for many of these draft reconstructions (since accurate data on this is limited), meaning gap-filling will add a lot of the same reactions to fulfill the exact biomass requirements.

6)Page 8, line 23 "Interestingly, several metabolic enzymes were consistently predicted to be necessary for observed metabolic capabilities (Figure 5C) or growth across all parasites (Table 1)"

-I am trying to understand how Figure 5C supports this claim along with Table 1. But it seems that Table 1 is missing. I can only find a legend for it: “Table 1: Most frequently gapfilled reactions. These reactions (in the BiGG namespace) were the most commonly added reactions as a result of all gapfilling steps.”

Either table 1 missing, or does it refer to supplementary table 1, and do we have to parse the data from there to understand what is claimed here?

7)"While reaction essentiality is generally more similar for closely related organisms (i.e. within genera), essentiality predictions were more similiar when comparing Plasmodium genomes to one another than between Toxoplasma or Cryptosporidium genomes.”

-I am confused here. Aren’t the results showing that as expected (“generally”) essentiality results are more similar for closely related Plasmodium genomes. I guess my confusion stems from the phrase starting with “while”, I assumed that the conclusion would be otherwise to what “generally” happens. So it’s either that or I am interpreting Figure 6 wrong.

I want to compliment the authors on clearly explaining the critical limitations of their pipeline in the discussion session. That addressed multiple questions I was going to include in my review.

**Have the authors made all data and (if applicable) computational code underlying the findings in their manuscript fully available?**

Reviewer #1: Yes

Reviewer #2: Yes

Reviewer #3: Yes

PLOS authors have the option to publish the peer review history of their article (what does this mean?). If published, this will include your full peer review and any attached files.

Reviewer #1: **Yes: **Mariana Galvao Ferrarini

Reviewer #2: No

Reviewer #3: No

Figure Files:

Data Requirements:

Reproducibility:

References:

---

## [Decision Letter · Decision Letter 1]

27 Jan 2022

Dear Ms. Carey,

We are pleased to inform you that your manuscript 'Comparative analyses of parasites with a comprehensive database of genome-scale metabolic models' has been provisionally accepted for publication in PLOS Computational Biology.

Best regards,

Pedro Mendes, PhD

Associate Editor

PLOS Computational Biology

Kiran Patil

Deputy Editor

PLOS Computational Biology

Reviewer's Responses to Questions

**Comments to the Authors:**

Reviewer #1: As I previously mentioned, the paper is very well written, the methods used are solid and the results reported are of extreme importance for the scientific community.

I also did not find the missing end of the sentence I mention in the previous review. It must have been my error when reading the first time around.

I fully support the acceptance of this paper and I am pleased with the answers to my questions and with the changes made to the original manuscript. I also congratulate the authors for a very nice read. 

On a side note: I have recently come across a novel model for T. cruzi which might be valuable for the refinement efforts of the authors in the future (in case the authors haven't seen it yet):

https://pubmed.ncbi.nlm.nih.gov/33021977/

Reviewer #2: The authors have revised the text thoroughly and provided all the information I asked for. I have no further comments, Congratulations!

Reviewer #3: Thank you for answering my questions and make the necessary changes to the manuscript.

**Have the authors made all data and (if applicable) computational code underlying the findings in their manuscript fully available?**

Reviewer #1: Yes

Reviewer #2: Yes

Reviewer #3: Yes

PLOS authors have the option to publish the peer review history of their article (what does this mean?). If published, this will include your full peer review and any attached files.

Reviewer #1: **Yes: **Mariana G. Ferrarini

Reviewer #2: No

Reviewer #3: No

---

## [Editor Report · Acceptance letter]

14 Feb 2022

PCOMPBIOL-D-21-01649R1 

Comparative analyses of parasites with a comprehensive database of genome-scale metabolic models

Dear Dr Carey,

I am pleased to inform you that your manuscript has been formally accepted for publication in PLOS Computational Biology. Your manuscript is now with our production department and you will be notified of the publication date in due course.

With kind regards,

Zsanett Szabo
